# Interventions to reintroduce or increase assisted vaginal births: a systematic review of the literature

Maria Regina Torloni [ID],[1] Newton Opiyo [ID],[2] Elena Altieri,[3] Soha Sobhy,[4] Shakila Thangaratinam,[5] Barbara Nolens [ID],[6] Diederike Geelhoed,[7] Ana Pilar Betran [ID] [2]

For numbered affiliations see end of article.

**Correspondence to**
Dr Maria Regina Torloni;
torlonimr@gmail.com

## ABSTRACT

**Objective** To synthesise the evidence from studies that implemented interventions to increase/reintroduce the use of assisted vaginal births (AVB).

**Design** Systematic review.

**Eligibility criteria** We included experimental, semi-experimental and observational studies that reported any intervention to reintroduce/increase AVB use.

**Data sources** We searched PubMed, EMBASE, CINAHL, LILACS, Scopus, Cochrane, WHO Library, Web of Science, ClinicalTrials.gov and WHO.int/ictrp through September 2021.

**Risk of bias** For trials, we used the Cochrane Effective Practice and Organisation of Care tool; for other designs we used Risk of Bias for Non-Randomised Studies of Interventions.

**Data extraction and synthesis** Due to heterogeneity in interventions, we did not conduct meta-analyses. We present data descriptively, grouping studies according to settings: high-income countries (HICs) or low/middle-income countries (LMICs). We classified direction of intervention effects as (a) statistically significant increase or decrease, (b) no statistically significant change or (c) statistical significance not reported in primary study. We provide qualitative syntheses of the main barriers and enablers for success of the intervention.

**Results** We included 16 studies (10 from LMICs), mostly of low or moderate methodological quality, which described interventions with various components (eg, didactic sessions, simulation, hands-on training, guidelines, audit/feedback). All HICs studies described isolated initiatives to increase AVB use; 9/10 LMIC studies tested initiatives to increase AVB use as part of larger multicomponent interventions to improve maternal/perinatal healthcare. No study assessed women's views or designed interventions using behavioural theories. Overall, interventions were less successful in LMICs than in HICs. Increase in AVB use was not associated with significant increase in adverse maternal or perinatal outcomes. The main barriers to the successful implementation of the initiatives were related to staff and hospital environment.

**Conclusions** There is insufficient evidence to indicate which intervention, or combination of interventions, is more effective to safely increase AVB use. More research is needed, especially in LMICs, including studies that design interventions taking into account theories of behaviour change.

## STRENGTHS AND LIMITATIONS OF THIS STUDY

⇒ This is the first systematic review on initiatives to increase the use of assisted vaginal birth, a life-saving procedure that has been steadily declining, especially in low-income and middle-income countries.

⇒ We conducted a comprehensive literature search (including grey literature), without date, publication status or language restrictions, in 10 electronic databases.

⇒ The systematic review followed strict methodological standards including double data extraction and quality assessment of all studies.

⇒ Many of the included studies had limited information on participant characteristics, lacked adequate statistical analyses and did not report important outcomes.

⇒ We created a subjective classification to categorise the core components of the interventions.

**PROSPERO registration number** CRD42020215224.

## INTRODUCTION

Assisted vaginal birth (AVB) refers to a vaginal birth conducted with the help of an instrument, such as a forceps or a vacuum extractor (VE). Common indications for AVB include prolonged second stage, maternal exhaustion or medical indications (eg, maternal heart disease), and suspected or confirmed fetal compromise in the second stage of labour.[1] AVB can reduce maternal and perinatal mortality and morbidity associated with complications in the second stage of labour.[2] AVB can also avoid a second-stage caesarean section (CS) and its associated short-term and long-term morbidity for the mother and baby, especially in low/middle-income countries (LMICs).[3–9] Compared with second-stage CS, AVB is associated with lower risks of haemorrhage, postoperative infection, faster maternal recovery, shorter hospital stay and cost-savings for health systems.[10–13] Moreover, women are more likely to have a spontaneous

vaginal birth in their next pregnancy after an AVB than after a CS.[14]

The benefits associated with AVB may be greater in low-resource settings where many births occur in primary care facilities with limited access to emergency obstetric care (EmOC), and where the need for a CS requires a time-consuming referral to a higher level health facility, where the safety of a CS cannot be guaranteed.[13 15–17] In these settings, AVB can be a life-saving intervention for mothers and babies who are facing second stage complications. The WHO and other UN agencies have included AVB as one of the seven critical signal functions of basic EmOC, and one of the nine signal functions of comprehensive EmOC.[18] However, reports indicate that, in most LMICs, AVB is the least likely function to be performed in primary care facilities.[19–21]

While CS rates have continued to increase in the last decades, rates of AVB have declined, particularly in LMICs.[22 23] Although unreliable data about type of birth may affect the accuracy of estimates especially in low-income settings, the prevalence of AVB in high-income countries (HICs) is currently 5%–20%, compared with <5% in most LMICs, with many countries having <1% of all women giving birth by AVB.[13] The reasons for the decline of AVB vary across settings and include the increasing use of epidurals for labour analgesia, lack of skilled operators and functioning equipment, concerns about trauma to the neonate and mother, fear of mother-to-child transmission of HIV, fear of litigation in case of complications and policies that exclude some professional cadres (eg, midwives) from performing AVB.[2 13 24 25]

In recent years, several investigators have reported efforts to increase or reintroduce AVB, particularly in LMICs, using a variety of strategies targeted at perceived barriers to the use of this procedure.[26–31] However, to our knowledge, there are no systematic reviews on this topic.

The general objective of this review was to identify, critically appraise and synthesise the evidence from studies which implemented interventions or programmes to increase or reintroduce AVB use. The specific objectives of the review were (a) to assess the effectiveness and safety of strategies aimed at increasing or reintroducing AVB use, (b) to identify the core elements of interventions to increase AVB use and (c) to describe the main barriers and enablers to the implementation of these initiatives, according to study authors.

## METHODS
The protocol for this review was registered in PROSPERO.[32] We report the review according to Preferred Reporting Items for Systematic Reviews and Meta-Analyses guidelines.[33]

### Patient and public involvement
Patients and the public were not involved in the design, or conduct, or reporting, or dissemination plans of our systematic review.

### Types of studies
We included studies that assessed the effects (benefits and harms) of interventions to reintroduce or increase AVB use with any of the following designs: randomised, semi-randomised, cluster randomised, stepped wedge, or non-randomised trials, cohort studies, cross-sectional studies, before and after intervention studies, interrupted time-series, large scale intervention case studies (implementation research studies) and mixed-methods (quantitative and qualitative) studies. Studies conducted in all settings (LMICs and HICs) were eligible for inclusion. We excluded studies presented only as congress abstracts as these have insufficient information about the intervention and/or outcomes.

### Participants
Studies involving health personnel who provide labour and delivery care to pregnant women (including qualified or trainee physicians, obstetricians, nurses, midwives and health assistants) were eligible for inclusion. We also included studies that implemented training programmes or quality improvement (QI) interventions to reintroduce or increase the use of AVB targeted at health systems, healthcare facilities or hospitals that provide labour and delivery care. We excluded studies that involved only undergraduate students.

### Interventions
Studies involving any intervention to introduce, reintroduce or increase AVB use with any instrument were eligible for inclusion. Educational or training programmes in any format and of any duration were eligible, including individual or group training, workshops or courses conducted online or face-to-face, onsite or offsite, involving simulation, hands-on training, and on the job supervision and auditing. We also included studies that used other strategies to increase AVB use, like different models of care, task-shifting, reorganisation of staff, use of opinion leaders or champions, and the provision or continued availability of functional equipment. We included studies that reported the use of specific, isolated interventions to increase AVB use as well as studies that promoted this intervention as part of larger educational/training or QI initiatives that included other components of EmOC. Eligible comparators were no intervention, usual care or practice in accordance with local protocols, or alternative interventions to reintroduce or increase the use of AVB, as reported in the primary studies.

### Outcome measures
The primary outcome was a change in the rate of AVB (overall or by specific instrument). Secondary outcomes were changes in CS rates (overall, intrapartum and second stage CS), maternal mortality and morbidity, neonatal mortality and morbidity, maternal birth experience, health provider satisfaction with training programme or QI intervention, and change in health provider AVB knowledge or skills. Studies that only reported secondary

outcomes without data on the primary outcome were not included.

We only included in the review studies that stated that an increase in AVB use was one of their objectives, or that change in AVB rate was one of the study outcomes (in the Methods section). All studies also had to provide data (before and after the intervention, or in the intervention and control groups) on AVBs performed by the study participants (Kirkpatrick level-3) or AVB prevalence in the hospital or setting (Kirkpatrick level-4).[34] We excluded studies that only provided data on participants' satisfaction with the intervention (Kirkpatrick level-1) or their knowledge acquisition (Kirkpatrick level-2).[34]

### Search strategy

We searched MEDLINE/PubMed, EMBASE, CINAHL, LILACS, Scopus, the Cochrane Library, WHO Library and Web of Science using the following combination of free text and MeSH terms adapted to each database: "forceps" OR "vacuum extraction" OR "ventouse" OR "operative vaginal delivery" OR "assisted vaginal birth" AND "training" OR "programs" OR "quality improvement initiatives" (complete search strategies in online supplemental file 1). We also searched MEDLINE/PubMed using the names of specific EmOC training programmes (ROBuST, ALSO, PRONTO, ESMOE). We searched two clinical trials registries (International Clinical Trials Registry Platform www.who.int/ictrp/en/ and ClinicalTrials.gov https://clinicaltrials.gov/) for ongoing studies or completed unpublished studies. We also searched websites of key organisations involved in EmOC training (WHO, UNFPA, UNICEF, JPHIEGO, FIGO, ICM, RCOG, Pathfinder, Population Council, Advanced Life Support Group and Childhealth Advocacy International). The searches were conducted in December 2019 and updated in September 2021. The searches were done without any language, date or publication status restrictions. We complemented the electronic searches by screening the reference lists of all studies selected for full-text reading and related reviews, and by contacting authors of relevant abstracts for additional, potentially relevant, studies.

### Process of study selection and data extraction

The identified references were entered in Covidence (https://www.covidence.org/). Four reviewers (MRT, NO, SS, APB) working in pairs independently screened titles and abstracts of all unique citations and applied the prespecified study eligibility criteria to select studies for full-text reading. Three independent reviewers (MRT, NO, APB) read these studies and selected those that fulfilled the aforementioned inclusion criteria. The same three reviewers extracted data on key information (study design, setting, participants, interventions, outcomes, and barriers and enablers to intervention success described by the authors) independently into a pilot-tested data extraction form specifically designed for this review. In all steps of the selection and extraction process, disagreements were solved by discussion.

In case consensus could not be reached, a fourth reviewer was called to arbitrate.

### Quality assessment of included studies

We used the criteria suggested by the Cochrane Effective Practice and Organisation of Care group to assess the risk of bias in the randomised trials included in the review.[35] Each trial was rated as having a 'high', 'low' or 'unclear' overall risks of bias based on nine domains: random sequence generation and allocation concealment (selection bias), baseline outcome measurements and baseline characteristics, blinding of participants and personnel (performance bias), blinding of outcome assessment (detection bias), contamination, incomplete outcome data (attrition bias) and selective reporting (reporting bias). For other study designs we used the 'Risk of Bias for Non-Randomised Studies of Interventions' tool.[36] Risk of bias domains in this tool include the likelihood of bias attributable to confounding, selection of participants, classification of interventions, deviations from intended interventions, missing data, measurement of outcomes and selection of reported results. Based on this approach, the overall risk of bias in each of these studies was rated as 'low risk', 'moderate risk', 'serious risk', 'critical risk' or 'no information'. Two review authors (NO, APB) independently conducted all quality assessments in duplicate. Disagreements were resolved by discussion, with the participation of a third reviewer when needed. We did not exclude studies based on their risk of bias ratings.

### Data synthesis

We intended to pool data from similar study designs into meta-analyses to assess the effectiveness of the intervention on the outcomes of interest. However, this was not possible because the interventions were very heterogeneous, with most individual studies using multiple different components in their educational and training interventions. Therefore, we present the data descriptively, grouping studies into two main categories according to the setting where they were conducted, that is, LMICs or HICs, according to the World Bank.[37] We classified the direction of effect of the intervention on each outcome as (a) statistically significant increase or decrease, (b) no statistically significant change or (c) statistical significance not reported by the primary study authors. Statistical significance was judged using standard statistics (p values or 95% CIs reported in primary studies). We list the main barriers and enablers related to the implementation or success of the intervention as reported by the study authors.

## RESULTS

We identified 13 219 unique records from electronic databases and clinical trials registries. We excluded 13 177 records following review of titles and abstracts, and retrieved the full texts of the remaining 42 records for detailed eligibility assessment; 32 of these records were

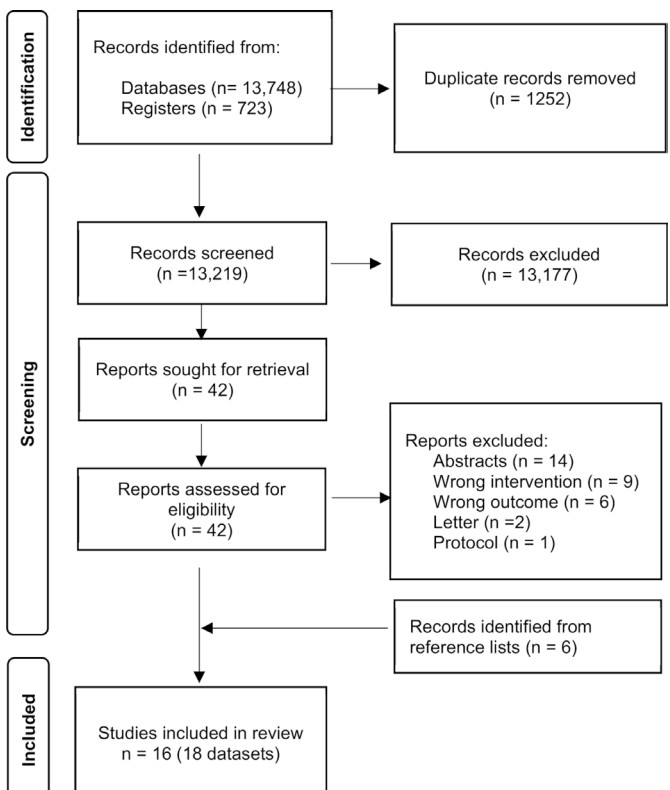

**Figure 1** Flowchart of process of study identification and inclusion.

excluded due to various reasons (online supplemental file 2). We included in the review six additional studies not captured by the electronic search that were identified from screening the reference lists of studies selected for full-text reading. Sixteen studies, fifteen published papers[26–31 38–46] and one PhD level thesis available online,[47] presenting eighteen datasets, fulfilled the selection criteria and were included in the review (figure 1).

### Main characteristics and quality of included studies

Six studies were conducted in three HICs (USA=4,[27 38 39 44] Australia=1,[43] Japan=1[45]). The other ten studies[26 28–31 40–42 46 47] were conducted in nine LMICs (Kenya, Mali, Mexico, Mozambique, Senegal, Tanzania, Thailand, Uganda and Ukraine). Most of these countries were in sub-Saharan Africa (Tanzania=3,[26 30 42] Kenya=1,[47] Mozambique=1,[31] Senegal and Mali=1,[40] Uganda=1[29]). Two studies[40 41] were conducted in two different LMICs and their results are presented in separate datasets. The other 14 studies were conducted in a single country. Most HIC studies (5/6) were conducted in single facilities while most LMIC studies (8/10) were multisite (ranging from 3 to 82 facilities). Most of the studies involved public facilities located in urban areas, included more than one type of healthcare professionals, and used initiatives to increase the use of VE, or forceps and VE. Midwives/nurses were not included in any of the HIC studies but participated in the AVB training interventions in all of the LMIC studies except one[29] (which focused exclusively on training obstetrics and gynaecology residents). The

studies were published mostly in 2010–2019 (11/16) and used non-randomised designs (13/16). Most studies were judged to have a moderate (8/16) or serious (3/16) risk of bias (table 1). Study details and quality assessments are provided in online supplemental files 3 and 4. Fourteen studies reported baseline rates of VE that ranged from 0% to 8.7%, seven studies reported baseline rates of forceps that ranged from 0% to 5.2%, and five studies presented baselines rates of AVB in general that ranged from 1.8% to 11.3%. Overall baseline CS rates were reported by 12 studies and ranged from 2.6% to 30.6%. In general, LMICs had the lowest and HICs had the highest baseline AVB and CS rates, respectively (online supplemental file 5).

### Core components of the interventions

Table 2 presents the main components of the interventions described in each study (details in online supplemental file 6). In almost all (9/10) of the studies conducted in LMICs, strategies to increase the use of AVB (mostly VE) were part of larger multicomponent interventions usually focused on improving EmOC, while all six studies from HICs tested interventions exclusively focused on increasing the use of forceps or both instruments. Didactic training (eg, lectures, theoretical classes) about AVB was part of the interventions in all studies conducted in LMICs, and in half of those conducted in HICs. The proportion of studies that included AVB simulation training was higher in LMIC (7/10) than in HIC studies (3/6). At least half of the studies conducted in both settings included short (up to 30 days) hands-on training periods when the healthcare professionals conducted AVB in women under the supervision of instructors. More prolonged (>30 days) on-site hands-on supervision was part of the intervention in most (4/6) of the studies conducted in HICs, but in less than half (4/10) of the LMIC studies. The provision of AVB guidelines and equipment was part of the intervention in about one-third (3/10) of the LMIC studies and in none of the HIC studies. Audit and feedback of AVB rates was a component of the intervention in almost half (4/10) of the LMIC studies and in only one of the HIC studies (table 2).

### Effects of the interventions

Table 3 presents the effects of the intervention on the primary outcome (AVB use) as described by the study authors. Only two of the studies[38 40] conducted analyses taking into account possible confounding factors that could have impacted the effects of the interventions on the rates of AVB (online supplemental file 3). In LMICs, the intervention was associated with a significant increase in VE or overall AVB rates in 5[29 40 42 46 47] of the 12 datasets from the 10 studies. Two additional studies[30 31] reported increases in VE rates but did not conduct statistical analyses. Therefore, in LMICs, interventions were associated with a successful effect (significant increase in AVB) in approximately 42% (5/12) of the datasets. Four of the six

**Table 1** Main characteristics of 16 studies on interventions to increase AVB use

| Characteristic | Studies (n) | References |
|---|---|---|
| Economic category (UN)* | | |
| Low income | 3 | Dumont et al (Mali), Geelhoed et al, Nolens et al[29 31 40] |
| Lower-middle income | 7 | Ameh et al, Berglund et al, Sequeira Dmello et al, Dumont et al (Senegal), Dominico et al, Mogilevkina et al, Sorensen et al[26 28 30 40 42 46 47] |
| Upper-middle income | 2 | Gülmezoglu et al[41] (Mexico and Thailand) |
| High income | 6 | Bardos et al, Becker et al, Cottrell et al, Skinner et al, Solt et al, Takeda and Ohashi[27 38 39 43–45] |
| Type of health facility† | | |
| Public | 12 | Ameh, Becker et al, Berglund et al, Sequeira Dmello et al, Dominico et al, Dumont et al, Geelhoed et al, Gülmezoglu et al (Thailand), Mogilevkina et al, Nolens et al, Skinner et al, Sorensen et al[26 28 30 31 39 40 42 43 46 47] |
| Private | 1 | Solt et al[44] |
| Public and private | 2 | Bardos et al, Gülmezoglu et al (Mexico)[38] |
| Unclear or not stated | 2 | Cottrell et al, Takeda and Ohashi[27 45] |
| Location of facility | | |
| Urban areas | 14 | Ameh, Bardos et al, Becker et al, Berglund et al, Cottrell et al, Sequeira Dmello et al, Dumont et al, Gülmezoglu et al, Mogilevkina et al, Nolens et al, Skinner et al, Solt et al, Sorensen et al, Takeda and Ohashi[26–29 38–40 42–47] |
| Rural areas | 1 | Geelhoed et al[31] |
| Urban and rural areas | 1 | Dominico et al[30] |
| Number of facilities | | |
| 1 | 7 | Bardos et al, Becker et al, Cottrell et al, Nolens et al, Solt et al, Sorensen et al, Takeda and Ohashi[26 27 29 38 39 44 45] |
| 3 | 2 | Berglund et al, Skinner et al[28 43] |
| 10–82 | 7 | Ameh, Sequeira Dmello et al, Dominico et al, Dumont et al, Geelhoed et al, Gülmezoglu et al, Mogilevkina et al[30 31 40 42 46 47] |
| Participants | | |
| Resident physicians | 4 | Bardos et al, Cottrell et al, Nolens et al, Solt et al[27 29 38 44] |
| Other physicians | 2 | Gülmezoglu et al, Takeda and Ohashi[45] |
| >1 type of professional‡ | 8 | Ameh, Becker et al, Sequeira Dmello et al, Dominico et al, Geelhoed et al, Mogilevkina et al, Skinner et al, Sorensen et al[26 30 31 39 42 43 46 47] |
| Unclear/not stated | 2 | Berglund et al, Dumont et al[28 40] |
| Type of instrument | | |
| Forceps | 2 | Bardos et al, Takeda and Ohashi[38 45] |
| Vacuum | 7 | Ameh, Sequeira Dmello et al, Dominico et al, Geelhoed et al, Gülmezoglu et al, Nolens et al, Sorensen et al[26 29–31 42 47] |
| Forceps and vacuum | 6 | Becker et al, Cottrell et al, Dumont et al, Mogilevkina et al, Skinner et al, Solt et al[27 39 40 43 44 46] |
| Unclear/not stated | 1 | Berglund et al[28] |
| Year of publication | | |
| 2020 or later | 4 | Becker et al, Cottrell et al, Sequeira Dmello et al, Mogilevkina et al[27 39 42 46] |
| 2010–2019 | 11 | Ameh, Bardos et al, Berglund et al, Dominico et al, Dumont et al, Geelhoed et al, Nolens et al, Skinner et al, Solt et al, Sorensen et al, Takeda and Ohashi[26 28–31 38 40 43–45 47] |
| 2000–2009 | 1 | Gülmezoglu et al |
| Study design | | |
| Randomised trials | 3 | Ameh, Dumont et al, Gülmezoglu et al[40 47] |

Continued

**Table 1** Continued

| Characteristic | Studies (n) | References |
|---|---|---|
| Before and after studies | 7 | Bardos *et al*, Berglund *et al*, Sequeira Dmello *et al*, Mogilevkina *et al*, Nolens *et al*, Sorensen *et al*, Takeda and Ohashi[26 28 29 38 42 45 46] |
| Interrupted time series | 3 | Dominico *et al*, Geelhoed *et al*, Skinner *et al*[30 31 43] |
| Cohort studies | 3 | Becker *et al*, Cottrell *et al*, Solt *et al*[27 39 44] |
| Risk of bias | | |
| Low | 4 | Ameh, Bardos *et al*, Dumont *et al*, Gülmezoglu *et al*[38 40 47] |
| Moderate | 8 | Berglund *et al*, Becker *et al*, Sequeira Dmello *et al*, Mogilevkina *et al*, Nolens *et al*, Skinner *et al*, Solt *et al*, Sorensen *et al*[26 28 29 39 42–44 46] |
| Serious | 3 | Cottrell *et al*, Dominico *et al*, Geelhoed *et al*[27 30 31] |
| Unclear§ | 1 | Takeda and Ohashi[45] |

*One study (Dumont *et al*) conducted in two countries: Mali (low-income) and Senegal (lower-middle income).
†One study (Gülmezoglu *et al*) conducted in public facilities (Thailand) and public and private facilities (Mexico).
‡Doctors and nurse/midwives (Ameh, Sequeira Dmello *et al*, Geelhoed *et al*, Mogilevkina *et al*); doctors, advance level associate clinicians and assistant medical officers (Dominico *et al*); obstetricians and residents (Becker *et al*, Skinner *et al*); nurse-midwives, assistant medical officers, doctors (Sorensen *et al*).
§Risk of bias assessment not done for one study (Takeda and Ohashi) due to insufficient information.
AVB, assisted vaginal birth.

studies conducted in HICs reported significant increases in forceps use associated with significant decreases in VE rates. The reasons for this effect are not described in the individual studies. One study conducted in Japan[45] also reported an increase in forceps and decrease in VE but did not provide statistical analyses, and one American study[38] reported a significant increase in forceps with non-significant change in VE rates. Since all studies in HICs focused mainly on increasing forceps use, we considered that the interventions were successful (significant increase) in 83% (5/6) of the studies conducted in these settings.

Many studies did not report any of the secondary outcomes of this review (online supplemental file 7). The effects of the interventions on the rates of CS were heterogeneous, ranging from significant decrease in overall CS rates,[38 40] to non-significant changes in overall CS rates[44 47] or CS for prolonged labour,[26] and significant increases in overall CS.[29 42 46] However, none of the 16 studies reported the effects of the intervention on second-stage CS rates (online supplemental files 7 and 8). In most of the studies that provided this information, significant increases in the rates of AVB were not associated with significant increases in adverse maternal or perinatal outcomes (online supplemental file 8). Out of the five studies[27 38 39 43 44] conducted in HIC countries with statistically significant increases in the use of forceps, one[39] did not assess any maternal or perinatal outcomes, and one[38] reported non-significant changes in adverse maternal and perinatal outcomes (3rd/4th degree tears, low 5 min Apgar scores). The other three HIC studies[27 43 44] reported that increased rates of forceps were associated with either non-significant changes in several adverse maternal or perinatal outcomes (3rd/4th degree tears, birth injuries,

overall neonatal complications, composite neonatal morbidity, low Apgar scores) or significant decreases in some of these outcomes (4th degree tears, overall maternal complications, postpartum haemorrhage, low cord pH) in this population. In all five studies conducted in LMIC with significant increases in AVB rates,[29 40 42 46 47] there were no significant increases in any of the adverse maternal outcomes reported, and several studies reported significant decreases in maternal mortality,[40 42] uterine rupture[29] and transfusions.[46] One LMIC study[46] did not report any perinatal outcomes, and two studies[40 47] reported either non-significant changes or significant decreases in adverse perinatal outcomes including stillbirth, neonatal death and admission to neonatal intensive care. The other two studies[29 42] had heterogeneous perinatal findings. One study conducted in Tanzania[42] reported that the increase in VE use was associated with a significant decrease in stillbirths along with a significant increase in neonatal deaths but this last outcome included both inborn and referred babies (who were born in other settings). A study conducted Uganda[29] reported that an increase in VE was associated with significant decreases in total perinatal mortality and intrapartum stillbirths, non-significant changes in neonatal deaths, and a significant increase in admissions of term infants to the neonatal intensive care unit (online supplemental file 8).

The two studies that assessed participants' changes in knowledge and skills (Kirkpatrick level-2) reported significant increases in these outcomes immediately after the training intervention.[39 47] Only one study assessed participants' satisfaction with the intervention (Kirkpatrick level-1) and reported that the trainees (mostly midwives) reacted positively to all lectures and breakout sessions

**Table 2** Main components of interventions to increase AVB use in 16 studies

| Study | Instrument of training | Focused only on AVB | Didactic training | Simulation training | Short* practical hands-on training | Long† onsite, hands-on supervision | AVB guidelines | Provision of equipment | Audit and feedback |
|---|---|---|---|---|---|---|---|---|---|
| 10 studies conducted in LMICs | | | | | | | | | |
| Ameh (Kenya)[47] | VE | No | Yes | Yes | Yes | Yes | No | Yes | No |
| Berglund et al (Ukraine)[28] | VE or F | No | Yes | No | No | No | Yes | No | Yes |
| Sequeira Dmello et al (Tanzania)[42] | VE | No | Yes | Yes | Yes | Yes | No | Yes | Yes |
| Dominico et al (Tanzania)[30] | VE | No | Yes | Yes | Yes | Yes | No | No | Yes |
| Dumont et al (Senegal and Mali)[40] | VE or F | No | Yes | Yes | No | No | Yes | No | No |
| Geelhoed et al (Mozambique)[31] | VE | No | Yes | No | Yes | No | No | No | Yes |
| Gülmezoglu et al (Mexico and Thailand) | VE | No | Yes | No | No | No | No | No | No |
| Mogilevkina et al (Ukraine)[46] | VE or F | No | Yes | Yes | No | No | No | No | No |
| Nolens et al (Uganda)[29] | VE | Yes | Yes | Yes | Yes | Yes | Yes | Yes | No |
| Sorensen et al (Tanzania)[26] | VE | No | Yes | Yes | No | No | No | No | No |
| 6 studies conducted in HICs | | | | | | | | | |
| Bardos et al (USA)[38] | F | Yes | No | No | Yes | Yes | No | No | No |
| Becker et al (USA)[39] | VE and F | Yes | Yes | Yes | No | No | No | No | No |
| Cottrell et al (USA)[27] | VE and F | Yes | No | No | Yes | Yes | No | No | No |
| Skinner et al (Australia)[43] | VE and F | Yes | Yes | Yes | Yes | Yes | No | No | Yes |
| Solt et al (USA)[44] | VE and F | Yes | No | No | Yes | Yes | No | No | No |
| Takeda and Ohashi (Japan)[45] | F | Yes | Yes | Yes | No | No | No | No | No |

*Short: up to 30 days.
†Long: More than 30 days.
AVB, assisted vaginal birth; F, forceps; HICs, high-income countries; LMICs, low/middle-income countries; VE, vacuum extraction.

(mean scores 9.3/10).[47] None of the studies assessed maternal birth experience or satisfaction during the interventions.

### Barriers and enablers to the success of the interventions

The main barriers to the success of interventions to increase AVB use were related to healthcare professionals, environmental factors and training. One of the main barriers reported by study authors was the difficulty in ensuring that the staff acquired and maintained all necessary skills to conduct AVB over time, including obstetrics clinical skills (correct diagnosis for the need to conduct an AVB), technical skills to safely use the instrument and non-technical skills (communication with the woman and team). Additional barriers were the lack of continued hands-on supervision and practice, lack of institutional support for instrumental births, lack of skilled personnel

(staff shortage, high turnover, need for retraining) and instruments, insufficient pain relief for women and fear of malpractice litigation in case of adverse outcomes. The main enablers to the success of the interventions included issues related to multidisciplinary training, clarity of the roles of healthcare providers, availability of trained and skilled healthcare professionals, and involving women in decision-making (table 4).

### DISCUSSION

We found relatively few studies, mostly conducted in LMICs, which reported interventions to increase the use of AVB. Due to the heterogeneity of the interventions, we could not conduct meta-analyses. In LMICs, most studies tested initiatives to increase AVB (mostly VE) as

**Table 3** Effectiveness of interventions to increase AVB use

| Study | Overall AVB rate | VE rate | Forceps rate |
|---|---|---|---|
| Low/middle-income countries (10 studies, 12 datasets*) | | | |
| Ameh (Kenya)[47] | | Increase† | |
| Berglund et al‡ (Ukraine) | NS change | | |
| Sequeira Dmello et al (Tanzania)[42] | | Increase† | |
| Dominico et al (Tanzania)[30] | | Increase | |
| Dumont et al (Mali)*[40] | NS change | | |
| Dumont et al (Senegal)*[40] | Increase* | | |
| Geelhoed et al (Mozambique)[31] | | Increase | |
| Gülmezoglu et al (Mexico)* | | NS change | |
| Gülmezoglu et al (Thailand)* | | NS change | |
| Mogilevkina et al (Ukraine)[46] | | Increase† | Decrease† |
| Nolens et al (Uganda)[29] | | Increase† | |
| Sorensen et al (Tanzania)[26] | | NS change | |
| High-income countries (six studies) | | | |
| Bardos et al (USA)[38] | | NS change | Increase† |
| Becker et al (USA)[39] | NS change | Decrease† | Increase† |
| Cottrell et al (USA)[27] | NS change | Decrease† | Increase† |
| Skinner et al (Australia)[43] | | Decrease† | Increase† |
| Solt et al (USA)[44] | NS change | Decrease† | Increase† |
| Takeda and Ohashi (Japan)[45] | | Decrease | Increase |

*Multi-country study with two separate datasets.
†Statistically significant.
‡Effect in two out of three sites.
AVB, assisted vaginal birth; NS, non-significant; VE, vacuum extraction.

part of larger multicomponent interventions focused on improving maternal/perinatal healthcare. On the other hand, the studies conducted in HICs involved stand-alone interventions focused specifically on increasing AVB use (mostly forceps). Effectiveness differed across studies but, overall, the interventions were less successful at increasing AVB use in LMICs (where baseline rates were lower) than in HICs. Increases in operative births were not associated with significant increase in adverse maternal or perinatal outcomes, in the studies that reported these outcomes. Caution is recommended in interpreting the findings of this review because most studies had moderate or serious risk of bias. The main barriers described by primary study authors to the success of the interventions were related to training and maintaining the skills of healthcare staff, lack of instruments and hospital environment factors.

Ensuring that participants acquired the necessary knowledge and skills to perform AVB was a core element of all interventions described in the studies. However, the diversity of methods, duration and combinations of intervention used in the 16 studies (eg, didactic training, simulation, hands-on supervision) precluded assessing the effectiveness of each component or combination. Although there seems to be consensus that it is essential to achieve and maintain the knowledge and skills of

the professionals caring for obstetric patients, the most effective way to do this remains unknown, particularly in LMICs.[48 49] However, to a certain extent, focusing exclusively on these aspects underestimates other behavioural factors, especially social and structural environmental factors, that need to be addressed for professionals to change their behaviour. Increasing the use of AVB is influenced by decision-making processes and involves behavioural change.[50] The COM-B model, one of the most widely used frameworks to understand behaviour, proposes that three essential conditions (capability, opportunity and motivation) interact to generate any behaviour.[51] Interventions that focus exclusively on healthcare providers' education and training (capability) may ignore other aspects, such as personal motivation and opportunity (social and physical environmental factors), that could play an important role in increasing the use of AVB. Systems thinking approaches may also help to identify behavioural and other drivers that can act as barriers.[52]

At least half of the studies involved other healthcare professionals besides doctors. As the main obstetric care providers in many LMICs, with appropriate training and patient selection, obstetric nurses and midwives can play an important role to increase or reintroduce AVB

**Table 4** Barriers and enablers to the implementation or success of interventions to increase AVB use

| Barrier categories | Barriers identified |
|---|---|
| Health professionals | Lack of skilled personnel (due to lack of training, high staff turnover, staff shortage)<br>Lack of confidence<br>Lack of supervision in general and especially in night shifts<br>Fear of HIV transmission<br>AVB perceived as a complicated and dangerous intervention<br>Resistance to change, previous attitudes and beliefs not supportive of AVB<br>Fear of malpractice litigation if complications arise |
| Environment | Lack of clarity on scope of midwifery practice and functions<br>Power struggles in the maternity units<br>Unsupportive work environment |
| Training | Lack of theoretical and formalised education on AVB<br>Need to learn skills for correct AVB indications and when to stop trying and do a CS<br>Lack of access to repeated opportunities to practice<br>Teacher/mentor preference influence the type of instrument to use and exposure to learning opportunities<br>Lack of access to or of willing clinical mentors<br>Poor communication with mentors |
| Women – Healthcare provider relationship | Insufficient pain relief for women<br>Lack of acceptance by the women (associated with negative interactions with staff, poor communication, little involvement in decision-making and mistrust of caregiver) |
| Equipment | Lack of equipment or maintenance of equipment<br>Difficulties related to the need to sterilise instruments (unavailability of the material/equipment for this purpose) |
| **Enabler categories** | **Enablers identified** |
| Health professionals | Senior midwifery support<br>Ownership of the initiative/strategy which is facilitated by Opinion Leaders<br>Retention of trained staff |
| Environment | Provision of opportunities for staff to gain experience<br>In areas where there is high population resistance to CS, AVB when indicated can be popular and acceptable<br>Clarity on the scope of the midwifery practice and functions<br>Establishment of task-shifting strategies<br>Involving senior staff and hospital administrators at the planning stage<br>Local stakeholders (Ministry of Health, hospital administrators) buy-in/endorsement<br>Advocacy efforts<br>Supportive environment including close supervision, proactive teaching by experienced colleagues enable the development of competencies |
| Training | Incorporating AVB training in the medical curriculum<br>Training using a range of tools and modalities including videos and simulation (these do not replace hands-on clinical teaching)<br>Training multidisciplinary teams<br>Retraining (in-service) |
| Women–healthcare provider relationship | Good communication between healthcare providers and with the woman to foster trust<br>Women's and partners' involvement in decision-making and satisfactory communication between women and providers<br>Acceptance of intervention by the woman is facilitated by positive interactions with staff, respectful care and ongoing communication and trust |
| Equipment | Availability of functioning, user friendly equipment |

AVB, assisted vaginal birth; CS, caesarean section.

use, especially in primary healthcare settings. Moreover, in LMICs, AVB in the hands of trained and competent midwives in primary care settings may be an important pathway to empower these professionals. In many LMIC peripheral settings, midwives are expected to manage a large number of births with little support or tools in case of complications, and limited possibilities for timely referral due to problems with transport, roads and distances.[31] Midwife-led models of care with backup from obstetricians, if and when necessary, could be promising

approaches to safely reintroduce AVB while ensuring sustainability.[53]

None of the studies reported the views of women, including their birth experience and satisfaction, during the interventions. This is an important dimension of healthcare evaluation since patient experience has been positively associated with clinical effectiveness and safety, and is a driver of healthcare provider decision-making.[54] Dignified and respectful care is an integral part of quality of care and especially relevant when managing complications or emergencies, such as the need for AVB, when women are often most vulnerable. Evidence shows that women recognise the unpredictable nature of birth but, when interventions are needed, they want to retain a sense of personal achievement and control through active decision-making.[55] The assessment of this dimension is important to address women's needs and fears related to AVB, which can directly affect the acceptability of this intervention.

Strengths of the review include its novelty (to our knowledge, this is the first systematic review on this topic), comprehensive literature search including grey literature sources, and adherence to rigorous methodological standards including double data extraction and quality assessment. Limitations of the review pertain mostly to the characteristics of the included studies, such as the paucity of studies that assessed relevant secondary outcomes (eg, second stage CS), or used adequate statistical analyses (eg, a priori power calculations or multivariate regression to control for confounders), and the limited information on participant characteristics. Additionally, we cannot exclude the possibility that our search may have missed relevant studies. Finally, we created a subjective classification to categorise the core components of the interventions and we acknowledge that this classification may not be ideal, and that there are other possible ways of doing this.

Based on the findings of this review, there does not seem to be a single recipe to successfully and safely increase AVB use. There is a need for well-designed, sufficiently powered experimental or implementation research studies, especially in LMICs. Ideally, future studies should design interventions after conducting local formative research to identify relevant barriers and facilitators for the desired behavioural change in the full range of relevant cadres and healthcare settings. Our findings suggest that, especially in studies conducted in LMICs, interventions should include a component of hands-on, onsite, prolonged supervision, as well as continuous training/retraining of new staff. Future studies should also assess and include women's perceptions and views on AVB in the design of the study, and evaluate participants' satisfaction with the initiative as one of their outcomes. Additionally, future studies should assess the effects of interventions on the rates of CS in the second stage of labour, and not only on the overall CS rate. Forceps and VE were the only instruments used in the included studies, with a clear predominance of VE in LMICs. The Odón device, a novel instrument that might be easier to use by all cadres of providers, is currently being evaluated for safety and feasibility.[56] This instrument could also be tested in future studies to increase AVD use. Finally, there is a need to assess the economic impact of strategies to increase AVB use, an important point especially for complex, multifaceted interventions that require substantial changes in organisations, staffing and model of care.

The identification of essential components or key features would be an important step to ensure more effective interventions. The use of Qualitative Comparative Analysis (QCA) is gaining popularity and could accelerate this process.[57] QCA is a methodology used for complex health interventions to identify if certain combinations of intervention features (eg, type of intervention, contextual characteristics and how the intervention was delivered) are associated with successful implementation and with improved uptake of the desired outcome, in this case, AVB.[57 58]

The review identified a paucity of studies that were grounded in behavioural theories. This contributed to the fragmented understanding of the specific influencing factors of the intended AVB behaviours, and the lack of identification of additional categories of barriers and drivers for the success of the intervention. Theories of behaviour and behaviour change provide guidance on such categories and their interaction which can help to understand the process of behaviour change before conducting any exploratory piloting and formal testing of interventions.[51 59]

## CONCLUSIONS

We identified relatively few, very heterogeneous, studies with multicomponent strategies to increase AVB use. The existing evidence is insufficient to indicate which intervention, or combination of interventions, is more effective to safely increase or reintroduce the use of AVB. More research is needed, in HICs as well as in LMICs, including studies that design interventions after formative research, take into account theories of behaviour change and systems thinking, and that assess all relevant outcomes.

**Author affiliations**

[1]Department of Medicine, Evidence Based Healthcare Post-Graduate Program, Sao Paulo Federal University, Sao Paulo, Brazil

[2]UNDP/UNFPA/UNICEF/WHO/World Bank Special Programme of Research, Development and Research Training in Human Reproduction (HRP) Department of Reproductive Health and Research, World Health Organization, Geneva, Switzerland

[3]Behavioral Insights Unit, World Health Organization, Geneva, Switzerland

[4]Women's Health Research Unit, Wolfson Institute of Population Health, Queen Mary University of London, London, UK

[5]Institute of Metabolism and Systems Research, University of Birmingham, Birmingham, UK

[6]Department of Obstetrics, Canisius Wilhelmina Hospital, Nijmegen, The Netherlands

[7]Provincial Directorate of Health, Tete Provincial Hospital, Cidade de Tete, Mozambique

**Contributors** All authors included on a paper fulfil the criteria of authorship. APB, MRT and NO conceived and planned the study, screened and selected the studies for inclusion, conducted data extraction and analyses, and wrote the first draft of

the study. EA, SS, ST, BN and DG made substantial contributions to data analysis and interpretation, and critically reviewed the manuscript for important intellectual content. MRT, NO, EA, SS, ST, BN, DG and APB approved the final version of the manuscript and agree to be accountable for all aspects of the work in ensuring that questions related to the accuracy or integrity of any part of the work are appropriately investigated and resolved. MRT is the guarantor.

**Funding** This research was funded by the UNDP-UNFPA-UNICEF-WHO- Bank Special Programme of Research, Development and Research Training in Human Reproduction (HRP), a cosponsored programme executed by the WHO in the Department of Sexual and Reproductive Health and Research.

**Competing interests** BN and DG were authors of studies included in the review but were not involved in the analyses of these studies. All authors have no other conflicts of interest to declare.

**Patient and public involvement** Patients and/or the public were not involved in the design, or conduct, or reporting, or dissemination plans of this research.

**Patient consent for publication** Not applicable.

**Ethics approval** This study does not involve human participants. Ethical approval was not required for this systematic review because all data came from information freely available in the public domain.

**Provenance and peer review** Not commissioned; externally peer reviewed.

**Data availability statement** All data relevant to the study are included in the article or uploaded as supplementary information.

**ORCID iDs**
Maria Regina Torloni http://orcid.org/0000-0003-4944-0720
Newton Opiyo http://orcid.org/0000-0003-2709-3609
Barbara Nolens http://orcid.org/0000-0001-9542-9392
Ana Pilar Betran http://orcid.org/0000-0002-5631-5883

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
