## [Reviewer comments · BMJ Open]

ARTICLE DETAILS

TITLE (PROVISIONAL)	Interventions to reintroduce or increase assisted vaginal births: a systematic review of the literature
AUTHORS	Torloni, Maria Regina; Opiyo, Newton; Altieri, Elena; Sobhy, Soha; Thangaratinam, Shakila; Nolens, Barbara; Geelhoed, D.; Betran, Ana Pilar

VERSION 1 – REVIEW

REVIEWER	Phung, Jason The University of Newcastle
REVIEW RETURNED	14-May-2022

GENERAL COMMENTS	Page 4, Line 29-32: unclear of what this sentence means. Page 4 Line 38-43: Authors should also consider the impact of unreliable documentation of birth type/databases, particularly in low income settings. Line 5-6; Page 6: Given the small number of studies, disagreements should be reviewed by an independent or 4th reviewer. Page 17, line 3: A very loaded statement that is not universal worldwide. I suggest this be rephrased. Page 17, Line 40-49: agree with the sentiment of this paragraph, but care should be taken as the goals of such planned research studies are different in HIC/LMIC countries. Page 17, Line 53-54: "This is not unexpected since vacuums are easier to use and perceived as more appropriate instruments for LMICs.": a rather subjective statement and should be rephrased. The focus of AVB is different in LMIC v.s. HIC. In LMIC, efforts to improve perinatal care are often assessed as education and training modules (which may include AVD) - this is reflected in most of your included studies which have not specifically addressed AVB per se, but overall intrapartum care and emergency training in obstetrics. As such, I don't think these interventions can adequately assess AVB in LMIC settings. An added complexity to these studies is the significant confounders of available resources in these study sites. In HIC, the push for increased AVB from my understanding is to try and reduce the often high CS rate (particularly, that in 2nd stage) - as this focus is entirely different, these two groups should not be compared in the same meta-analysis. I recommend that should the authors be interested in AVB specifically, LMIC and HIC need to be assessed separately with
--

	differing outcomes. HIC studies should focus on AVB rates and secondary outcomes: 2nd stage CS, maternal and neonatal outcomes. It is not well described why studies in HIC report increased forceps rates with a decreased vacuum rates. The LMIC studies are too heterogenous in their intervention, outcomes and study populations to attempt meta-analysis, even descriptively. A review article describing the interventions may be more appropriate in this setting.
--	--

REVIEWER	zimmo, kaled Aqsa Martyrss hospital, Obstetrics and Gynaecology
REVIEW RETURNED	16-May-2022

GENERAL COMMENTS	Thank you for the authors to focus on this important topic recently were highlighted as one of the denominators to reducing the CS globally. The authors presented and discussed the manuscript in a very impressive way. I have only some comments:  - Conclusion: the sentence stated that. More research is needed, especially in LMICs, I think not only in those countries but globally is an essential issue? - I think if the authors can add more details about the strength and limitations of the published study included, they will add more insight into the challenges. - Do references no 20,22,30 need to be revised well? - Table 2: is very impressive but table 3 needs to discuss the possible confounding factors for each intervention.
---

REVIEWER	Sandoval, Grecio The George Washington University Biostatistics Center
REVIEW RETURNED	14-Jul-2022

GENERAL COMMENTS	DATA SYNTHESIS: Please briefly explain and describe what the heterogeneity of the interventions means. Was it that different studies utilized different interventions or that individual studies utilized multiple different kinds of interventions? How were LMICs and HICs defined? What level of significance was used to define statistically significant? Was statistically significant study specific and just based on whether the study reported 'significance' or not? Were the levels of significance different for each study? Please clarify how 'statistically significant' was defined, the levels of significance that were used for each of the studies, and briefly describe what level of significance this systematic review considered to be 'statistically significant'. Of those studies that reported statistical significance, were these based on unadjusted or multivariable analysis? If there were multivariable analyses, which types of covariates did these study adjusted for? RESULTS: On page 6, Line 49, should there be a comma instead of decimal, for "13.177"? The additional studies that were not included from the electronic search seem arbitrary here. Please describe and clarify how it was decided that the "six additional studies not captured by the
---

electronic search” were included. For “one thesis available online”, please consider describing this as “PhD level thesis” in the text.

In this results section, please also report the number of datasets (versus studies) used.

MAIN CHARACTERISTICS AND QUALITY OF INCLUDED STUDIES: In this section of the text, the authors state “The other ten studies were conducted in nine LMICs”, but only 5 LMICs were listed and the sum of the counts amount to only 7 studies. Please clearly summarize and report the number of studies and number of countries included in each economic category.

For the two studies conducted in two countries, were the countries within those studies from the same economic category? Were the results from these countries pooled in those analyses or were separate analyses done for each of those countries? Please clarify.

Were the ranges for the rates of VE, forceps, AVB, and CS notably different between HICs and LMICs? It seems like this information would be important to be reported since your analysis was done by economic category.

CORE COMPONENTS OF THE INTERVENTIONS: The sentence on Page 9, Line 7 “In almost all (9/10) of the studies conducted LMICs, strategies to increase the use of AVB...” is not clear. Do you mean “...studies conducted from LMICs...”?

EFFECTS OF THE INTERVENTIONS: The result as stated on Page 11, Lines 13-15 is not completely true based on the data reported on Table 3, “Five of the 6 studies conducted in HICs reported significant increases in forceps use associated with significant decreases in VE rates.” This statement makes it sound like that those 5 studies also reported decreases in VE rates, which is not true based on Table 3 (one of them has “NS change”). Please review and reword this result appropriately.

Please clarify “...the rates of CS were heterogeneous”. Were they heterogeneous overall, or within each economic category (e.g. HICs and LMICs)?

TABLE 3: It looks like there were 10 studies with 12 datasets (instead of 11 datasets as labeled in the table). From the table, it looks like the Dumont 2013 study and Gulmezoglu 2006 study each have 2 datasets. Should this be 12 datasets?

BARRIERS AND ENABLERS TO THE SUCCESS OF THE INTERVENTIONS: How come the frequencies of these reported barriers are not reported? Did all studies report these barriers? Which barriers were the most commonly reported? Were the barriers reported from studies from HICs and LMICs similar?

FIGURE 1: In the ‘Included’ section, please consider also reporting the number of datasets included (in addition to the number of studies). For example, 16 studies included with ## datasets.

SUPPLEMENTARY FILE 3, DETAILS OF 16 STUDIES ON INTERVENTIONS TO INCREASE AVB USE: For studies that reported statistical analysis, if a confidence interval was not

	reported, please consider reporting the p-value (and level of significance). It is difficult to quickly review whether a result was significant without a confidence interval (or p-value). Please see the previous comment about the number of datasets. Based on this table it also looks like the Dumont 2013 study and Gulmezoglu 2006 study each have 2 datasets (each has two results reported). SUPPLEMENTARY FILE 8, SUMMARY OF EFFECTS OF THE INTERVENTIONS IN 16 STUDIES INCLUDED IN THE REVIEW: Please see previous comment for Supplementary File 3; for studies that reported statistical analysis, if a confidence was not reported, please report the p-value (and level of significance). For Ameh 2014, what does “21.5% x 21.5%” mean? For Becker 2020, what does the bolded 30.6% and 33.5% mean? For Berglund 2010, what does “difce” mean? For Dmello 2021, what does “formo” mean? For Moglievkina 2022, if there is a decrease in forceps rate from 0.17% to 0.08%, why is the point estimate odds ratio greater than 1 (OR=1.80)? For Takeda 2018, What does “1.2-13.3% to 2.3%” mean? Why are some numbers bolded and some are not bolded?
--	--

VERSION 1 – AUTHOR RESPONSE

Reviewer 1. Dr. Jason Phung, The University of Newcastle, Hunter Medical Research Institute

Comment	Reply
Page 4, Line 29-32: unclear of what this sentence means.	We rewrote this sentence to clarify its meaning
Page 4 Line 38-43: Authors should also consider the impact of unreliable documentation of birth type/databases, particularly in low income settings.	We added this consideration to the paragraph about the rates of AVB.
Line 5-6; Page 6: Given the small number of studies, disagreements should be reviewed by an independent or 4 th reviewer	This was the procedure that we followed. In case consensus could not be reached between the three reviewers, a fourth reviewer was called to arbitrate. We added this information to the revised manuscript.
Page 17, line 3: A very loaded statement that is not universal worldwide. I suggest this be rephrased.	We agree. We deleted this statement which refers to a qualitative study conducted in one province of Mozambique and thus it cannot be considered universal.
Page 17, Line 40-49: agree with the sentiment of this paragraph, but care should be taken as the goals of such	We have modified the sentence adding this consideration.

planned research studies are different in HIC/LMIC countries.	
Page 17, Line 53-54: "This is not unexpected since vacuums are easier to use and perceived as more appropriate instruments for LMICs.": a rather subjective statement and should be rephrased.	Agreed. We deleted this statement.
The focus of AVB is different in LMIC v.s. HIC. In LMIC, efforts to improve perinatal care are often assessed as education and training modules (which may include AVD) - this is reflected in most of your included studies which have not specifically addressed AVB per se, but overall intrapartum care and emergency training in obstetrics. As such, I don't think these interventions can adequately assess AVB in LMIC settings. An added complexity to these studies is the significant confounders of available resources in these study sites. In HIC, the push for increased AVB from my understanding is to try and reduce the often high CS rate (particularly, that in 2nd stage) - as this focus is entirely different, these two groups should not be compared in the same meta-analysis. I recommend that should the authors be interested in AVB specifically, LMIC and HIC need to be assessed separately with differing outcomes. HIC studies should focus on AVB rates and secondary outcomes: 2nd stage CS, maternal and neonatal outcomes.	We thank the reviewer for the reflection embedded in this comment showing his understanding of the complexity of this issue. We understand the reviewer's comment and agree that the motivation to perform (more) AVBs may differ in HIC compared to LMIC. Accordingly, we interpreted the findings of the review under the same perspective as the reviewer. Having said that, this systematic review did not aim to investigate the motivation that led the 1ary study authors to report interventions to increase AVB (although this is usually evident in the background section as the rationale for the study). The use of AVBs is declining worldwide, and this is considered a public health problem because AVB is an important obstetric intervention for some women who develop complications in the 2nd stage of labor. Therefore, as stated in the manuscript, the objectives of our systematic review were: a) to assess the effectiveness and safety of strategies aimed at increasing or reintroducing AVB use, b) to identify the core elements of interventions to increase AVB use, and c) to describe the main barriers and enablers to the implementation of these initiatives, according to study authors. We explain in Methods that "We included studies that reported the use of specific, isolated interventions to increase AVB use as well as studies that promoted this intervention as part of larger educational /training or QI initiatives that included other components of EmOC." We note the reviewer's comment that in HICs, the push for AVB is to try to reduce the high use of CS in 2nd stage of labor. Interestingly, this is also becoming increasingly a problem in some settings in LMICs. In many of these countries, there are large inequalities in access; underuse and overuse of CS coexist and CS are often unsafe [See references below]. In the last decade, governments and clinicians have expressed concern about the rise in CS and the potential negative consequences for maternal and infant health, particularly in LMICs. As with any surgery, CSs are associated with short and long-term risks that can extend many years after the index delivery and affect the health of the woman, her child, and future pregnancies. These risks are higher in women with limited access to comprehensive obstetric care. The emerging

problem of overuse of CS in LMICs may have triggered more recent research in LMICs to include the reduction of CS use as one of the explicit objectives/benefits of programs to improve perinatal care, with increasing focus on AVB training [See the studies captured in this review in Uganda (Nolens et al 2016); Mozambique (Geelhoed et al 2018); or Tanzania (Dominico et al 2018 and Dmello et al 2021)]. These are studies that specifically focused on AVB to reduce the use of potentially unnecessary and unsafe CS.

We agree with the reviewer that meta-analysis is not appropriate, and we did not conduct any meta-analyses in this systematic review. However, as suggested by the reviewer, we do report the characteristics of the interventions (Table 2) and the effectiveness of the interventions (Table 3) separately for studies conducted in HICs and in LMICs, so that the differences between these two groups of countries could be more evident visually. Aligned with the reviewer's comments, these differences underpin our results and discussion sections.

References;

- Betran AP, Torloni MR, Zhang JJ, Gülmezoglu AM; WHO Working Group on Caesarean Section. WHO Statement on Caesarean Section Rates. BJOG. 2016 Apr;123(5):667-70.
- Boatin AA, Schlottheuber A, Betran AP, Moller AB, Barros AJD, Boerma T, Torloni MR, Victora CG, Hosseinpoor AR. Within country inequalities in caesarean section rates: observational study of 72 low and middle income countries. BMJ. 2018 Jan 24;360:k55.
- Betran AP, Ye J, Moller A-B, et al. Trends and projections of caesarean section rates: global and regional estimates. BMJ Global Health 2021;6:e005671.
- Betrán AP, Temmerman M, Kingdon C, Mohiddin A, Opiyo N, Torloni MR, Zhang J, Musana O, Wanyonyi SZ, Gülmezoglu AM, Downe S. Interventions to reduce unnecessary caesarean sections in healthy women and babies. Lancet. 2018 Oct 13;392(10155):1358-1368.
- Fawcus S, Pattinson RC, Moodley J, Moran NF, Schoon MG, Mhlanga RE, Baloyi S, Bekker E, Gebhardt GS. Maternal deaths from bleeding associated with caesarean delivery: A national emergency. S Afr Med J. 2016 Apr 7;106(5):53-7.
- Victora CG, Barros FC. Beware: unnecessary caesarean sections may be hazardous. Lancet. 2006;367(9525):1796-7.

	 • Lumbiganon P, Laopaiboon M, Gulmezoglu AM, Souza JP, Taneepanichskul S, Ruyan P, et al. Method of delivery and pregnancy outcomes in Asia: the WHO global survey on maternal and perinatal health 2007-08. Lancet. 2010;375:490-9. • Souza JP, Gulmezoglu A, Lumbiganon P, Laopaiboon M, Carroli G, Fawole B, et al. Caesarean section without medical indications is associated with an • Increased risk of adverse short-term maternal outcomes: the 2004-2008 WHO Global Survey on Maternal and Perinatal Health. BMC medicine. 2010;8:71.
It is not well described why studies in HIC report increased forceps rates with a decreased vacuum rates.	Two of the six studies conducted in HICs (Bardos 2017 and Takeda 2018) presented interventions involving only training/education to increase the use of forceps. The other four studies involved interventions to increase the use of both instruments, but resulted in significant increases in forceps rates, associated with significant decreases in VE rates. The reasons for this effect are not described in detail in the individual studies; therefore, we can only hypothesize about the possible reasons underpinning this result. We added a sentence to explain these findings in the Results section (under Effects of the Intervention).
The LMIC studies are too heterogenous in their intervention, outcomes and study populations to attempt meta-analysis, even descriptively. A review article describing the interventions may be more appropriate in this setting.	Heterogeneity does not preclude evidence synthesis. In fact, heterogeneity is an outcome/finding in itself. We agree that the LMIC as well as the HIC studies included in this systematic review were heterogeneous in their interventions. As stated in Methods (under Data synthesis), we did not pool data from individual studies into meta-analyses to assess the effectiveness of the interventions to increase AVB use because “most individual studies used multiple different components”. Therefore, we present the results of the studies narratively, as recommended by guidelines for the conduct of systematic reviews of effectiveness (See: McKenzie JE, Brennan SE. Chapter 12: Synthesizing and presenting findings using other methods. In: Higgins JPT, Thomas J, Chandler J, Cumpston M, Li T, Page MJ, Welch VA (editors). Cochrane Handbook for Systematic Reviews of Interventions version 6.3 (updated February 2022). Cochrane, 2022). The 1ary outcome of this systematic review (changes in AVB rates after the intervention) was reported by all 16 studies; therefore, there was no heterogeneity in our 1ary outcome.

	The type of healthcare providers (population) included in the studies is inevitably heterogeneous because AVB can be performed by diverse cadres (e.g., physicians, midwives, nurses, health assistants) in different settings, and in each country or setting, the relevant cadres need to be trained. It is important to reconcile research methods with the emerging challenges of societies and health systems. Our systematic review, as many contemporary reviews involving complex issues, collected real-world experiences which are heterogeneous by nature in some aspects. Therefore, as suggested by the reviewer, we describe our findings narratively.
--	---

Reviewer 2.

Dr. kaled zimmo, Aqsa Martyr hospital, intervention center, Oslo university , Rikshospitalet

Thank you for the authors to focus on this important topic recently were highlighted as one of the denominators to reducing the CS globally. The authors presented and discussed the manuscript in a very impressive way.	Thank you for your comment. We are delighted that the reviewer considers this an important topic, and that he values the efforts and the way the manuscript is presented and discussed. A committed multidisciplinary team of professionals made this review possible.
I have only some comments:- Conclusion: the sentence stated that. More research is needed, especially in LMICs, I think not only in those countries but globally is an essential issue?	We agree and modified the sentence to reflect the need for research in HICs as well as in LMICs.
I think if the authors can add more details about the strength and limitations of the published study included, they will add more insight into the challenges.	We agree but, since the word count of the manuscript is already above the 4000 limit, we cannot comply with this suggestion. Study details and quality assessments are provided in Supplementary files 3 and 4.
Do references no 20,22,30 need to be revised well?-	References 20, 22 and 30 in the list at the end of the manuscript are formatted according with the style recommended by BMJ Open. In Pubmed, these references also appear exactly as in our manuscript. 20. Ameh C, Msuya S, Hofman J, et al. Status of emergency obstetric care in six developing countries five years before the MDG targets for maternal and newborn health. PLoS One 2012;7(12):e49938. doi: 10.1371/journal.pone.0049938 22. Betran AP, Ye J, Moller AB, et al. The Increasing Trend in Caesarean Section Rates: Global, Regional and National Estimates: 1990-2014. PLoS One 2016;11(2):e0148343. doi: 10.1371/journal.pone.0148343 30. Dominico S, Bailey PE, Mwakatundu N, et al. Reintroducing vacuum extraction in primary health care

	facilities: a case study from Tanzania. BMC Pregnancy Childbirth 2018;18(1):248. doi: 10.1186/s12884-018-1888-9
Table 2: is very impressive but table 3 needs to discuss the possible confounding factors for each intervention.	Only two of the included studies (Bardos 2017 and Dummont 2013) conducted analyses taking into account possible confounding factors that could have impacted the effects of the interventions on the rates of AVB. We added this information in Results, Effects of the interventions. Details of the variables included in the adjusted analyses were added to Supplementary file 3.

Reviewer: 3 Dr. Greccio Sandoval, The George Washington University Biostatistics Center

DATA SYNTHESIS: Please briefly explain and describe what the heterogeneity of the interventions means. Was it that different studies utilized different interventions or that individual studies utilized multiple different kinds of interventions?	To clarify the point raised by the reviewer, we added a sentence explaining that “most individual studies used multiple different components in their educational and training interventions”.
How were LMICs and HICs defined?	We added that the countries were classified according to the World Bank system and a new reference (number 37).
What level of significance was used to define statistically significant? Was statistically significant study specific and just based on whether the study reported ‘significance’ or not? Were the levels of significance different for each study? Please clarify how ‘statistically significant’ was defined, the levels of significance that were used for each of the studies, and briefly describe what level of significance this systematic review considered to be ‘statistically significant’	We used the level of statistical significance reported by the 1ary study authors. Therefore, as you state, “statistically significant was study specific and just based on whether the study reported ‘significance’ or not”. The level of significance used by each study (when available) was added to Supplementary Table 3, in the second column (Study design, Year study conducted).
Of those studies that reported statistical significance, were these based on unadjusted or multivariable analysis? If there were multivariable analyses, which types of covariates did these study adjusted for?	We added this information to Supplementary table 3 (Details of the studies), in the last column (Effects of the Intervention).
RESULTS: On page 6, Line 49, should there be a comma instead of decimal, for “13.177”?	Yes, this should be a comma. We corrected it.
The additional studies that were not included from the electronic search seem arbitrary here. Please describe and clarify how it was decided that the “six additional studies not captured by	The identification of all potentially relevant studies is recognized as a critical component of the systematic review process. To reduce the risk of missing studies not identified through electronic searches, systematic review guidelines recommend screening the reference lists of all

the electronic search” were included. For “one thesis available online”, please consider describing this as “PhD level thesis” in the text.	studies selected for full-text reading. We used this process (as described in Methods), and that is how we identified the additional studies mentioned by the reviewer. We reworded the sentence in Results to clarify this point. We added “PhD level thesis”, as suggested.
In this results section, please also report the number of datasets (versus studies) used.	We added this information to the first paragraph of Results and to Figure 1.
MAIN CHARACTERISTICS AND QUALITY OF INCLUDED STUDIES: In this section of the text, the authors state “The other ten studies were conducted in nine LMICs”, but only 5 LMICs were listed and the sum of the counts amount to only 7 studies. Please clearly summarize and report the number of studies and number of countries included in each economic category.	We rephrased this sentence, citing the nine individual LMICs, to clarify the information.
For the two studies conducted in two countries, were the countries within those studies from the same economic category? Were the results from these countries pooled in those analyses or were separate analyses done for each of those countries? Please clarify.	The two multicountry studies were conducted in two different LMICs. The results of these multicountry studies were analyzed separately in the original articles and the authors of these studies present independent results for each country. Accordingly, we extracted the data for each country separately to avoid masking differences between countries typical of summary measures. We added this information to Results, under Main characteristics and quality of included studies.
Were the ranges for the rates of VE, forceps, AVB, and CS notably different between HICs and LMICs? It seems like this information would be important to be reported since your analysis was done by economic category.	We did not conduct statistical analyses to compare the baseline rates of VE, forceps, AVB and CS reported by the individual studies from different settings. However, we added the name of the country next to the study reference in Supplementary file 5 (Baseline rates of AVB and CSs in the included studies). In addition, we added a sentence in Results (under Main characteristics and quality of included studies) mentioning that, in general, LMICs had the lowest and HICs had the highest baseline AVB and CS rates, respectively.
CORE COMPONENTS OF THE INTERVENTIONS: The sentence on Page 9, Line 7 “In almost all (9/10) of the studies conducted LMICs, strategies to increase the use of AVB...” is not clear. Do you mean “...studies conducted from LMICs...”?	Yes, that is exactly what we meant. We corrected the sentence for clarity.
EFFECTS OF THE INTERVENTIONS: The result as stated on Page 11, Lines 13-15 is not	We corrected the number of studies in the sentence to reflect the exact findings reported on Table 3.

completely true based on the data reported on Table 3, “Five of the 6 studies conducted in HICs reported significant increases in forceps use associated with significant decreases in VE rates.” This statement makes it sound like that those 5 studies also reported decreases in VE rates, which is not true based on Table 3 (one of them has “NS change”). Please review and reword this result appropriately	
Please clarify “...the rates of CS were heterogeneous”. Were they heterogeneous overall, or within each economic category (e.g. HICs and LMICs)?	The effects of the interventions on the rates of CS were heterogeneous overall. We added this word to the sentence to improve clarity.
TABLE 3: It looks like there were 10 studies with 12 datasets (instead of 11 datasets as labeled in the table). From the table, it looks like the Dumont 2013 study and Gulmezoglu 2006 study each have 2 datasets. Should this be 12 datasets?	The reviewer is correct, and we are grateful for pointing this typo. We corrected the number of datasets (from 11 to 12) in the heading of Table 3 for the first group of studies.
BARRIERS AND ENABLERS TO THE SUCCESS OF THE INTERVENTIONS: How come the frequencies of these reported barriers are not reported? Did all studies report these barriers? Which barriers were the most commonly reported? Were the barriers reported from studies from HICs and LMICs similar?	Table 4 lists the barriers and enablers to the success of the intervention descriptively. We did not set out to quantify how many times each barrier and enabler was mentioned in each of the included study. The assessment of barriers and facilitators was not a formal objective in any of the 16 studies included in this systematic review. Therefore, these studies did not use rigorous quantitative (e.g., surveys) or qualitative (e.g., in depth interviews) methods to assess barriers and enablers. These were mentioned in the Discussion sections of the studies, as comments by the investigators who participated in the field work of these studies. Therefore, we used an informal approach to compile this information from the 16 studies. The objective of this part of our systematic review was to maximize the use of the information provided in the original articles and point to possible future studies.
FIGURE 1: In the ‘Included’ section, please consider also reporting the number of datasets included (in addition to the number of studies). For example, 16 studies included with ## datasets.	As requested by the reviewer, we added this information to Figure 1.
SUPPLEMENTARY FILE 3, DETAILS OF 16 STUDIES ON INTERVENTIONS TO INCREASE AVB USE: For studies that reported statistical analysis, if a confidence interval was not reported, please	We added the exact p-values for the differences that were statistically significant to Supplementary table 3 (Details of the studies), in the last column (Effects of the Intervention).

consider reporting the p-value (and level of significance). It is difficult to quickly review whether a result was significant without a confidence interval (or p-value). Please see the previous comment about the number of datasets. Based on this table it also looks like the Dumont2013 study and Gulmezoglu 2006 study each have 2 datasets (each has two results reported).	Yes, Dumont 2013 and Gulmezoglu 2006 have 2 datasets each. See also response above, related to the studies with datasets for more than one country.
SUPPLEMENTARY FILE 8, SUMMARY OF EFFECTS OF THE INTERVENTIONS IN 16 STUDIES INCLUDED IN THE REVIEW: Please see previous comment for Supplementary File 3; for studies that reported statistical analysis, if a confidence was not reported, please report the p-value (and level of significance)	We added this information to Supplementary file 8.
For Ameh 2014, what does “21.5% x 21.5%” mean?	All percentages indicate rates (of AVB or CS) or incidence of outcomes at baseline x after the intervention. This explanation was added at the top of Supplementary file 8, for clarity.
For Becker 2020, what does the bolded 30.6% and 33.5% mean?	Bold terms in the cells of the table indicate rates or outcomes with statistically significant changes after the intervention (compared to baseline) according to primary study authors. This explanation was added at the top of Supplementary file 8, for clarity.
For Berglund 2010, what does “difce” mean?	This is a typing error. We corrected the word to “difference”.
For Dmello 2021, what does “formo” mean?	This is a typing error. We corrected the word to “from”.
For Moglievkina 2022, if there is a decrease in forceps rate from 0.17% to 0.08%, why is the point estimate odds ratio greater than 1 (OR=1.80)?	We thank the reviewer for the careful reading of the results of this study. Moglievkina et al. evaluated the impact of the intervention using a Difference-in-differences (DID) analysis to compare outcome changes over time between the intervention and control groups and then calculated the ratio of the OR of the DID. The DID shows a non-statistically significant difference in change in the use of forceps during the period studied between the intervention and the control group (DID=0.07 CI, 0.00-0.13). The ratio of OR reported of 1.80 (CI, 1.00-3.25) represents the ratio of two odds ratios: the odds ratio between periods (before/after) in the use of forceps in controls and the odds ratio between periods in the use of forceps in the intervention group. Values greater than 1 indicate that the change in the use of forceps in controls is larger than the change in the use of forceps the

	intervention group. This result corresponds with the DID estimator where the reduction in forceps use was larger in the control than in the intervention group. In other words, the use of forceps decreased in both groups: control and intervention but it decreased more in the control group, although the difference was not statistically significant. We have added this information to Supplementary File 8 to better describe this.
For Takeda 2018, What does “1.2-13.3% to 2.3%” mean? Why are some numbers bolded and some are not bolded?	In the 10 pre-intervention years (baseline), the rates of AVB ranged from 1.2% -13.2%; after the intervention, the rate of AVB was 2.3%. We added this information to Supplementary File 8. All numbers are now presented as not bolded.

VERSION 2 – REVIEW

REVIEWER	Sandoval, Grecio The George Washington University Biostatistics Center
REVIEW RETURNED	04-Dec-2022

GENERAL COMMENTS	GENERAL COMMENTS: This is essentially a resubmission of the manuscript BMJOPEN-2022-062986. From a statistical perspective, the authors provided a reasonable explanation for not performing a formal meta-analysis as the interventions across the included studies were widely varied (i.e. typical meta-analyses would combine studies with the same intervention or treatment). In this analysis, the authors opted to summarize results descriptively or in a narrative manner; however some statements in the results could be better supported by providing a quantified result (e.g. frequency or count). The authors have addressed most of my previous comments, but there are still some results that need to be clarified. RESULTS, MAIN CHARACTERISTICS AND QUALITY OF INCLUDED STUDIES: Page 7, Line 23: Text reported “CS rates were reported by five studies...” but Supplementary File 5 reports 12 studies. Page 11, Line 27-29: It might be helpful to also include the range of the effect size values (stratified by studies that reported overall cesarean vs. cesarean in the second stage of labour) to exemplify heterogeneity. Page 11, Line 30-32: If this is true, then the statement may best be supported and presented along with a quantified result (e.g. provide the number of studies that resulted in significant differences)? I do not think Supplementary File 8 supports the statement “most of the interventions to increase AVB use were not associated with significant changes in maternal morbidity or perinatal morbimortality.” It looks like most of the studies resulted in statistical differences in maternal and perinatal outcomes. SUPPLEMENTARY FILE 3:
---

	Thank you for the explanation about the use of Difference-in-Differences and ratio of odds ratios. However upon reviewing this reference, it looks like the Mogilevkina et al. (https://link.springer.com/article/10.1186/s12884-022-04458-9/tables/3) reported a statistically significant increase (marginally) in forceps, but Table 3 and Supplementary File 3 reports “No significant change in Forcep rates” for this study.
--	--

REVIEWER	zimmo, kaled Aqsa Martyrss hospital, Obstetrics and Gynaecology
REVIEW RETURNED	20-Dec-2022

GENERAL COMMENTS	The authors responded to my comments adequately thank you
---

VERSION 2 – AUTHOR RESPONSE

Reviewer 1. Dr. Grecio Sandoval, The George Washington University Biostatistics Center

Comment	Reply
GENERAL COMMENTS: This is essentially a resubmission of the manuscript BMJOPEN-2022-062986. From a statistical perspective, the authors provided a reasonable explanation for not performing a formal meta-analysis as the interventions across the included studies were widely varied (i.e. typical meta-analyses would combine studies with the same intervention or treatment). In this analysis, the authors opted to summarize results descriptively or in a narrative manner; however some statements in the results could be better supported by providing a quantified result (e.g. frequency or count). The authors have addressed most of my previous comments, but there are still some results that need to be clarified.	Yes, this is a resubmission of manuscript BMJOPEN-2022-062986. We are grateful for the numerous comments and suggestions that you made to the first version of the manuscript.
RESULTS, MAIN CHARACTERISTICS AND QUALITY OF INCLUDED STUDIES: Page 7, Line 23: Text reported “CS rates were reported by five studies...” but Supplementary File 5 reports 12 studies. Page 11, Line 27-29: It might be helpful to also include the range of the effect size values (stratified by studies that reported overall cesarean vs. cesarean in the second stage of labour) to exemplify heterogeneity. Page 11, Line 30-32: If this is true, then the statement may best be supported and presented	We corrected the information in the text of the manuscript. None of the studies reported second stage CS. We have added a sentence about the

along with a quantified result (e.g. provide the number of studies that resulted in significant differences)? I do not think Supplementary File 8 supports the statement “most of the interventions to increase AVB use were not associated with significant changes in maternal morbidity or perinatal morbimortality.” It looks like most of the studies resulted in statistical differences in maternal and perinatal outcomes.

heterogeneous effects of the interventions on the rates of CS.

Our concern was that increases in AVB would be associated with **increases in adverse maternal and perinatal outcomes**. However, as can be seen in Supplementary File 8, in most of the studies with significant increase in AVB, this did not occur.

To clarify this point, in the text of the manuscript we changed the sentence that you mentioned to “In most of the studies that provided this information, significant increases in the rates of AVB were not associated with significant **increases in adverse maternal or perinatal outcomes**.”

Following your suggestion, and to further clarify this point, we added the following sentences to this section of the manuscript: “Out of the five studies^{27,38,39,43,44} conducted in HIC countries with statistically significant increases in the use of forceps, one³⁹ did not assess any maternal or perinatal outcomes, and one³⁸ reported non-significant changes in adverse maternal and perinatal outcomes (3rd/4th degree tears, low 5-minute Apgar scores). The other three HIC studies^{27,43,44} reported that increased rates of forceps were associated with either non-significant changes in several adverse maternal or perinatal outcomes (3rd/4th degree tears, birth injuries, overall neonatal complications, composite neonatal morbidity, low Apgar scores) or significant decreases in some of these outcomes (4th degree tears, overall maternal complications, postpartum haemorrhage, low cord pH) in this population. In all five studies conducted in LMIC with significant increases in AVB rates^{29, 40, 42, 46 47}, there were no significant increases in any of the adverse maternal outcomes reported, and several studies reported significant decreases in maternal mortality^{40,42}, uterine rupture²⁹, and transfusions⁴⁶. One LMIC study⁴⁶ did not report any perinatal outcomes, and two

	studies^{47,40} reported either non-significant changes or significant decreases in adverse perinatal outcomes including stillbirth, neonatal death, and admission to neonatal intensive care. The other two studies^{42,29} had heterogeneous perinatal findings. One study conducted in Tanzania⁴² reported that the increase in VE use was associated with a significant decrease in stillbirths along with a significant increase in neonatal deaths but this last outcome included both inborn and referred babies (who were born in other settings). A study conducted Uganda²⁹ reported that increase in VE was associated with significant decreases in total perinatal mortality and intrapartum stillbirths, non-significant changes in neonatal deaths, and a significant increase in admissions of term infants to the neonatal intensive care unit. “ We also changed the wording of the Abstract (last sentence of Results) to clarify this point from “Increase in AVB use was not associated with significant change in adverse maternal or perinatal outcomes” to “Increase in AVB use was not associated with significant increase in adverse maternal or perinatal outcomes”. Finally, in Supplementary File 8 we changed the title of the last column from “Changes in maternal & perinatal outcome” to “Changes in adverse maternal & perinatal outcomes”. And we added highlights in different colors in the last column to easily identify changes in adverse maternal and perinatal outcomes that were non-significant (yellow), that decreased (green), or that increased (grey).
SUPPLEMENTARY FILE 3: Thank you for the explanation about the use of Difference-in-Differences and ratio of odds ratios. However upon reviewing this reference, it looks like the Mogilevkina et al. (https://link.springer.com/article/10.1186/s12884-022-04458-9/tables/3) reported a statistically significant increase (marginally) in forceps, but	Sorry for the error: the change in the rate of forceps in the intervention group was indeed statistically significant. However, the rate of forceps actually decreased in the period after the intervention*. Therefore, we corrected this information in Table 3, Supplementary File 3, and Supplementary File 8 of our systematic

Table 3 and Supplementary File 3 reports “No significant change in Forcep rates” for this study.	review. In Supplementary File 3, we also added the P values for the DID estimator presented in Mogilevkina’s Table 3. *In the text of the Abstract (and some other parts of the manuscript), Mogilevkina et al state that there was a significant “increase” in the use of forceps after the intervention. This is incorrect, as can be seen in Table 3, which provides numerical data. The rate of forceps in the intervention group (called “cases” by the authors) actually decreased from 0.17% (80/47,838) in the before period to 0.08% (47/61,116) in the after period. This also occurred in the control group (called “referent” by the authors), where the rate of forceps declined from 0.21% (80/38,032) in the before period to 0.05% (23/42,866) in the after period. In the 2nd paragraph of Results, the authors (correctly) state that the “Use of forceps assisted delivery declined in both groups, but only for the cases was that decrease significant”. This information is also presented in Table 2 and Fig 2 of their paper.
---	--

Reviewer 2. Dr. kaled zimmo, Aqsa Martyr hospital, intervention center, Oslo university, Rikshospitalet

Comment	Reply
The authors responded to my comments adequately thank you	Thank you.